# Genomic Prediction of Resistance to Tan Spot, Spot Blotch and Septoria Nodorum Blotch in Synthetic Hexaploid Wheat

**DOI:** 10.3390/ijms241310506

**Published:** 2023-06-22

**Authors:** Guillermo García-Barrios, José Crossa, Serafín Cruz-Izquierdo, Víctor Heber Aguilar-Rincón, J. Sergio Sandoval-Islas, Tarsicio Corona-Torres, Nerida Lozano-Ramírez, Susanne Dreisigacker, Xinyao He, Pawan Kumar Singh, Rosa Angela Pacheco-Gil

**Affiliations:** 1Postgrado en Recursos Genéticos y Productividad-Genética, Colegio de Postgraduados, Campus Montecillo, Texcoco 56264, Estado de México, Mexico; garcia.guillermo@colpos.mx (G.G.-B.); sercruz@colpos.mx (S.C.-I.); aheber@colpos.mx (V.H.A.-R.); tcoronat@colpos.mx (T.C.-T.); 2International Maize and Wheat Improvement Center (CIMMYT), Km 35 Carretera México-Veracruz, Texcoco 56237, Estado de México, Mexico; n.lozano@cgiar.org (N.L.-R.); s.dreisigacker@cgiar.org (S.D.); x.he@cgiar.org (X.H.); pk.singh@cgiar.org (P.K.S.); 3Postgrado en Socioeconomía, Estadística e Informática, Colegio de Postgraduados, Campus Montecillo, Texcoco 56264, Estado de México, Mexico; 4Postgrado en Fitosanidad, Colegio de Postgraduados, Campus Montecillo, Texcoco 56264, Estado de México, Mexico; sandoval@colpos.mx

**Keywords:** breeding values, wheat diseases, genomic selection, pedigree and genomic relationship matrices

## Abstract

Genomic prediction combines molecular and phenotypic data in a training population to predict the breeding values of individuals that have only been genotyped. The use of genomic information in breeding programs helps to increase the frequency of favorable alleles in the populations of interest. This study evaluated the performance of BLUP (Best Linear Unbiased Prediction) in predicting resistance to tan spot, spot blotch and Septoria nodorum blotch in synthetic hexaploid wheat. BLUP was implemented in single-trait and multi-trait models with three variations: (1) the pedigree relationship matrix (A-BLUP), (2) the genomic relationship matrix (G-BLUP), and (3) a combination of the two matrices (A+G BLUP). In all three diseases, the A-BLUP model had a lower performance, and the G-BLUP and A+G BLUP were statistically similar (*p* ≥ 0.05). The prediction accuracy with the single trait was statistically similar (*p* ≥ 0.05) to the multi-trait accuracy, possibly due to the low correlation of severity between the diseases.

## 1. Introduction

Breeding for resistance to diseases is a central focus in plant breeding programs, since any successful variety must have the complete package of a high grain yield, resistance to diseases, desirable agronomic characteristics, and end-use quality [1].

Tan spot, spot blotch and Septoria nodorum blotch are leaf-spotting diseases in wheat with global importance. The necrotrophic fungus *Pyrenophora tritici-repentis* (Died.) Drechsler causes tan spot and occurs frequently in Canada, the USA, Australia and South Africa. Spot blotch is caused by *Bipolaris sorokiniana* Shoemaker and causes intense damage in the wheat-growing areas of South Asia, Latin America and southern Africa [2]. The filamentous ascomycete *Stagonospora nodorum* Berkeley is the causal agent of the Septoria nodorum blotch, which is of economic importance in Australia, the USA, Western Europe and southern Brazil [2,3].

Conventional plant breeding has created wheat lines with major disease resistance genes (qualitative resistance), and minor Quantitative Trait Loci (QTL) are usually neglected, which restricts the diversity of resistance genes in new varieties, and a substantial part of the genetic variance provided by the minor-effect loci is lost [4,5].

A promising approach in plant breeding for disease resistance is genomic selection, which can select both major and minor QTLs that synergistically confer more durable resistance. When the major QTLs are rendered ineffective, the quantitative resistance (provided by minor QTLs) will still provide a good level of protection against a pathogen population boom [1].

Genomic selection uses a “training population” (which comprises individuals that have been genotyped and phenotyped) to train a model that predicts the breeding values of individuals in a selection population that have not been phenotyped [6].

Genomic prediction can be applied to single traits and multiple traits. Single-trait models predict the value of a single phenotype in a test data set, whereas multi-trait models predict two or more traits simultaneously. In general, multi-trait models represent complex relations between traits more efficiently since they use correlations between genotypes and traits [7].

The accuracy of genomic prediction is affected by several genetic factors, including marker density, training set size, the relationship between the training population and test population, population structure, the heritability and genetic architecture of target traits, and linkage disequilibrium between markers and causal variants. In addition to the above genetic factors, the statistical method is another important factor affecting the predictive ability [8].

Synthetic hexaploid wheats (SHWs) provide an option in the search for genetic resistance to diseases. This wheat is generated by hybridizing modern durum wheat (*Triticum turgidum* L.; 2n = 4x = 28, AABB) and wild goat grass (*Aegilops tauschii* Coss.; 2n = 2x = 14, DD), the F_1_ triploid of which is added colchicine to induce the duplication of chromosomes [9,10]. The sub-genome D in SHW has a higher genetic diversity than the sub-genome D in bread wheat [10,11]. This diversity is a resource used to identify resistance to various diseases [12,13,14]. There are studies of genomic prediction in SHW for traits such as grain yield [15,16] and yellow rust [17], yet to the best of our knowledge, this is the first report on genomic prediction for tan spot, spot blotch and Septoria nodorum blotch.

The aim of this study was to compare the predictive ability of three prediction models that use (1) the pedigree relationship matrix (A-BLUP), (2) the genomic relationship matrix (G-BLUP) and (3) the combination of these two matrices (A+G BLUP). The models were implemented as both single-trait and multi-trait models to predict resistance to tan spot, spot blotch and Septoria nodorum blotch.

## 2. Results

### 2.1. Analysis of Phenotypic Data

The Best Linear Unbiased Estimator (BLUE) for severity had a mean of 1.63 for tan spot, 1.64 for spot blotch and 1.61 for Septoria nodorum blotch. The minimum values were similar for the three diseases, around 0.95, whereas the maximum values were 3.47, 4.08 and 3.67 for tan spot, spot blotch and Septoria nodorum blotch, respectively. For the three diseases, the standard error of the mean was low, and heritability (H^2^) was relatively high, between 0.84 and 0.89, while the genomic heritability estimates ranged from 0.54 to 0.68 (Table 1). Some cultivars with missing values are adjusted when BLUEs are computed, so that the minimum values appear lower than 1 (Table 1).

The values of the correlations, although positive and significant (*p* < 0.01), were low: r = 0.27 between Septoria nodorum blotch and spot blotch, r = 0.39 between tan spot and spot blotch and a value between Septoria nodorum blotch and tan spot (r = 0.46) that was considered moderate.

### 2.2. Analysis of Genotypic Data

Genotyping via sequencing produced 6548 single-nucleotide polymorphisms (SNPs), of which 1616 (24.6%) belonged to sub-genome A, 1653 (25.2%) to sub-genome B and 3279 (50%) to sub-genome D. After purifying and imputing the markers, 4053 SNPs remained useful.

### 2.3. Prediction Accuracy for Seedling Resistance to Tan Spot, Spot Blotch and Septoria Nodorum Blotch

The most accurately predicted disease was tan spot in both the single-trait and multi-trait models. The prediction accuracy fluctuated between 0.67 (single-trait G-BLUP and A+G BLUP models) and 0.39 (multi-trait A-BLUP model) (Table 2, Figure 1). The mean square error (MSE) was low for the six models, being between 0.135 and 0.224.

For all three diseases, the pedigree model (A-BLUP) had a lower performance than the other two, in both the single=trait and multi-trait cases, and at the same time, this model was different (*p* < 0.01) from the G-BLUP and A+G BLUP models (Figure 1, Figure 2 and Figure 3).

For spot blotch, the model with the lowest prediction accuracy was the A-BLUP multi-trait model (r = 0.27 and MSE = 0.260), and the one with the highest performance was the A+G BLUP multi-trait model (r = 0.45 and MSE = 0.225), which represents an advantage of 66.6% in terms of prediction accuracy (Table 3). The single-trait models had a similar prediction accuracy to the multi-trait models (*p* ≥ 0.05) (Figure 2).

The accuracy for Septoria nodorum blotch fluctuated between 0.40 (single-trait A-BLUP) and 0.55 (multi-trait G-BLUP and A+G BLUP) (Table 2). Comparing the single-trait and multi-trait implementations shows that the three multi-trait models had an advantage in accuracy and a slight reduction in the MSE, although these differences were not significant (Figure 3).

### 2.4. Impact of Marker Density

When evaluating different marker densities, it was found that the prediction accuracy increased with a higher marker density, although the increase in accuracy became marginal beyond 4000 markers. A similar response was observed for all three diseases. The mean accuracy ranged from 0.87 to 0.90 for tan spot, 0.79 to 0.84 for spot blotch and from 0.86 to 0.91 for Septoria nodorum blotch (Figure 4).

## 3. Discussion

SHWs are a source of genetic diversity that can be used in breeding programs for diverse traits. This study found a high proportion (50%) of SNPs in the sub-genome D of SHW, which contrasts with the results obtained by Phuke et al. [18], who found a frequency of 7.5%, and Juliana et al. [19] (between 7.4 and 10%) for the same sub-genome in bread wheat.

The diversity of sub-genome D and the high resistance to these three diseases indicate that SHWs are useful parents for developing varieties. However, SHWs usually house unfavorable alleles; therefore, complementary strategies must be followed in breeding programs, such as backcrossing with adapted elite lines [15].

### 3.1. Prediction Accuracy for Tan Spot

This study obtained a prediction accuracy ranging from 0.39 to 0.67 for this disease; these values are higher than those obtained by Semagn et al. [20], who obtained accuracies between 0.11 and 0.41 in a diverse collection of spring bread wheats. These same authors found higher accuracies (up to 0.75) in populations of recombinant endogamous lines when using models with genomic and environmental information.

Another study in which predictions were carried out for tan spot in bread wheat, both in seedling and in adult plants, obtained accuracies of up to 0.77 for seedlings and 0.57 for adult plants. The best model included information on the markers and pedigree [19]. In this regard, several studies have shown that genomic data or a combination of genomic data and pedigree data surpass the models that only include pedigree [21,22,23].

### 3.2. Prediction Accuracy for Spot Blotch

In this case, a maximum accuracy of 0.45 was obtained using the A+G BLUP model, which is lower than that reported by Juliana et al. [24] for bread wheat collections, where an accuracy of 0.53 was obtained using the G-BLUP model.

### 3.3. Prediction Accuracy for Septoria Nodorum Blotch

Regarding the predictions for this disease, Juliana et al. [19] obtained values from 0.43 to 0.63 in bread wheat. The model with the best performance included genomic and pedigree information. In this study, the models with the best performance (G-BLUP and A+G BLUP) had an accuracy of 0.55; although this is a lower precision, the studies coincide in demonstrating that the most precise models include genomic and pedigree information.

### 3.4. Models with Genomic Information

The good performance of the G-BLUP and A+G BLUP models in this study can partially be explained by the fact that the genomic information helps to predict the Mendelian sampling, and it has the ability to use the information of close and distant relatives (as long as the markers are in a linkage imbalance with the causal loci) [25].

No differences were found in the prediction accuracies of the G-BLUP and A+G BLUP models (*p* ≥ 0.05), and similar results were reported by Crossa et al. [21], who stated that the advantage of considering the pedigree and markers jointly can be small in some cases due to the redundance between the regression of the pedigree and the regression of the markers.

### 3.5. Single-Trait vs. Multi-Trait Models

The multi-trait models were expected to have a better prediction accuracy than the single-trait models, but this was not the case in the current study. This is due to the low to moderate correlations between the severity of the diseases.

Although all three pathogens had ToxA toxin, it is known that these pathogens harbor other toxin-producing genes that could have contributed to the low correlations. When the correlation between traits is low, the correlation matrix rarely has useful information that one can incorporate into multi-trait prediction models [26].

### 3.6. Marker Density

All three traits showed the highest prediction accuracy when all the available markers were used. However, beyond 4000 markers, the improvement in prediction accuracy was minor, showing that this was sufficient for generating a relatively accurate prediction calibration within this panel.

Prediction accuracy increases with marker density; however, this increase will eventually reach a plateau. Norman et al. [27] evaluated different marker densities to predict grain yield, thousand kernel weight, glaucousness, and maturity in a panel of bread wheats, finding that with 5000 markers, a relatively accurate prediction calibration can be generated.

The threshold at which plateau occurs is determined by the extent of linkage disequilibrium between markers and causal variants, the genetic architecture of the trait and heritability. Therefore, a panel with high linkage disequilibrium and traits with high heritability may have a reduced number of markers, with less impact on accuracy [27,28].

## 4. Materials and Methods

### 4.1. Plant Material

Four hundred SHWs, developed using the CIMMYT (International Maize and Wheat Improvement Center, Texcoco, Mexico) Wide-Cross Program, were used. These genotypes were selected from a group of 1524 lines with favorable agronomic characteristics and resistance to several diseases.

### 4.2. Inoculation and Evaluation of Disease Severity

The three diseases were evaluated in the seedling stage (when the blade of the second leaf was fully expanded). The experiments were carried out independently for each disease between 2018 and 2019 in greenhouses at the experimental station located in El Batán, Edo. de Mexico, Mexico (altitude 2240 m, 19.5° N, 98.8° W).

The experiments were organized in a randomized complete block design, with 12 replicates for tan spot and Septoria nodorum blotch and six for spot blotch. Four seedlings were used for every repetition, all planted in plastic trays. Fungal isolates that produce ToxA were used for the three diseases (Table 3). The inoculation details can be found in [14,29].

The severity of the diseases was evaluated seven days after inoculation, according to the Lamari and Bernier ordinal scale, ranging from 1 to 5 [30]. Based on the average of the repetitions, the genotypes were grouped into resistant (1.0 to 1.5), moderately resistant (1.6 to 2.5), moderately susceptible (2.6 to 3.5) and susceptible (3.6 to 5.0) groups.

### 4.3. Genotyping

The genomic DNA was taken from 10-day-old seedlings using the cetyltrimethyl-ammonium bromide method in a 96-well plate format [31].

The DArT genotyping, based on sequencing, followed the DArTseq™ methods (Canberra, Australia). DNA was digested with the restriction enzymes PstI (5′-CTGCA|G-3′) and HpaII (5′-CCG|G-3′) and barcode adaptors were linked to individual samples.

The libraries were sequenced using Illumina Novaseq 6000 (Illumina, CA, USA) in the lab of the Servicio de Análisis Genético para la Agricultura (Genetic Analysis Service for Agriculture—SAGA) in CIMMYT, Mexico. SNP calling was carried out with the analytical pipeline patented by DArT P/L (Canberra, Australia).

The SNP marker matrix was converted into the 1, 0, −1 format, where it is 1 if the individual is homozygous for the most frequent allele, 0 if it is heterozygous and −1 if it is homozygous for the least frequent allele.

The SNP markers were purified when eliminating markers with more than 40% of the data missing and with a minor allele frequency below 5%. The missing data were imputed using the expectation–maximization algorithm.

### 4.4. Analysis of Phenotypic Data

With the data for the severity of each disease, the BLUE was calculated along with the broad-sense heritability (H^2^) using the software META-R version 6.0 [32]. Later, the mean, standard deviation, median, minimum, maximum and the standard error of the mean were also obtained. The Pearson correlations between the three diseases were obtained using the psych package version 2.2.9 in R [33]. Genomic heritability was calculated with a linear regression of the markers.

### 4.5. Genomic Prediction Models and Cross-Validation Scheme

The breeding values were predicted using the BLUP (Best Linear Unbiased Prediction) model, in the single-trait and multi-trait cases, with three variants: (1) using the pedigree relationship matrix (A-BLUP), (2) using the genomic relationship matrix (G-BLUP) and (3) combining the two matrices (A+G BLUP).

The simple main effects model A-BLUP (also known as the infinitesimal model) was implemented as follows:(1)$$y_{j}=\mu +a_{j}+e_{j}$$
where $y_{j}$ is the response of the *j*th wheat line, μ is the intercept, $a_{j}$ is an approximation of the true genetic value of the *j*th line derived from the calculation of the additive relationship matrix of the pedigree (coefficient of parentage A), and $e_{j}$ is the residual, for which is assumed that $e_{j}\overset{iid}{\sim }N(0,\sigma_{e}^{2})$. Here, N represents a random variable that is normally distributed, and “iid” represents a variable that is independently and identically distributed.

Other assumptions of the model are that the vector of random effects $a_{j}$ follows a normal, multivariate density with a mean of 0 and a variance–covariance matrix $A\sigma_{a}^{2}$, $a\sim N\left(0,A\sigma_{a}^{2}\right),$ where $\sigma_{a}^{2}$ is the infinitesimal component of additive variance. The random effects $\alpha =\left(a_{1},\ldots ,a_{J}\right)\prime$ are correlated in such a way that the model allows for the borrowing of information through the wheat lines; therefore, prediction is possible.

G-BLUP is similar to Model 1, except for the fact that random effect $a_{j}$ is replaced with $g_{j}$, which is an approximation of the real genetic value of the *j*th wheat line. This approach is implemented via the regression of the markers SNP $g_{j}=\sum_{m=1}^{p}x_{jm}b_{m}$, where $x_{jm}$ is the genotype of the *j*th line in the *m*th marker, and $b_{m}$ is the effect of the *m*th marker with the assumption that $b_{m}\overset{iid}{\sim }N(0,\sigma_{b}^{2})(m=1,\ldots ,p)$, where $\sigma_{b}^{2}$ is the variance in the effects of the markers.

Vector $g=\left(g_{1},\ldots ,g_{J}\right)\prime$ contains the genomic values of all the lines, and it is assumed to follow a normal multivariant density with a mean of zero and a covariance matrix $Cov\left(g\right)=G\sigma_{g}^{2}$, where **G** is the genomic relationship matrix and $\sigma_{g}^{2}$, which is proportional to $\sigma_{b}^{2}$ ($\sigma_{g}^{2}\propto \sigma_{b}^{2}$), is the genomic variance. Therefore, Model 1 becomes:(2)$$y_{j}=\mu +g_{j}+e_{j}$$
where the random effects vectors are assumed to be $g\sim N(0,G\sigma_{g}^{2})$ and $e_{j}\overset{iid}{\sim }N(0,\sigma_{e}^{2})$. The random effects $g=\left(g_{1},\ldots ,g_{J}\right)\prime$ are correlated in such a way that Model 2 allows for the borrowing of information through the wheat lines, therefore making prediction possible. As indicated, $g_{j}$ comes near to the true genetic values of line $L_{j}$ [34].

**G** is a matrix of genomic relations given by $G_{ij}=\frac{\sum_{m=1}^{p}\left(x_{im}-2p_{m}\right)(x_{jm}-2p_{m})}{\sum_{m=1}^{p}{2p}_{m}(1-p_{m})}$, where $p_{m}$ is the estimated allelic frequency, with a number of copies in the *i*th individual that is counted in $x_{im}$. When centering (that is, when subtracting $2p_{m}$ from the genotypes) or standardizing (that is, dividing by $\sum_{m=1}^{p}2p_{m}(1-p_{m})$), it is possible to interpret $\sigma_{g}^{2}=\sigma_{b}^{2}\sum_{m=1}^{p}2p_{m}(1-p_{m})$ as a genomic variance. As the number of independent segregation loci increases, the entries of the genomic relationship matrix **G** converge to twice the coefficient of parentage (or coancestry) between lines (**A**).

The model A+G BLUP is constructed when joining pedigree Model 1 and genomic Model 2 as:(3)$$y_{j}=\mu +g_{j}+a_{j}+\varepsilon_{j}$$
of which the components were described earlier.

The multi-trait models were implemented according to Pérez and de los Campos [35]:(4)$$\left[\begin{array}{c} \begin{array}{c} y_{1}\\ y_{2} \end{array}\\ y_{3} \end{array}\right]=\left[\begin{array}{c} \mu_{1}\\ \mu_{2}\\ \mu_{3} \end{array}\right]+\left[\begin{array}{c} {x_{j}^{t}\beta }_{1}\\ {x_{j}^{t}\beta }_{2}\\ {x_{j}^{t}\beta }_{3} \end{array}\right]+\left[\begin{array}{c} e_{1}\\ e_{2}\\ e_{3} \end{array}\right]$$
where $y_{ij}$  *i* = 1, 2, 3, *j* = 1, …, *n* is the response of the *i*th disease in the *j*th wheat line; $\mu_{1},\mu_{2},\mu_{3}$ are the intercepts for each disease; x is an incidence matrix for the effects of a set of predictors (in our case, markers or pedigree); β is a matrix of the effects of a set of predictors; and $e_{1},e_{2},e_{3}$ denote the random residue vector.

It is assumed that $\left(e_{1},e_{2},e_{3})\sim MVN(O,I⊗R\right)$, where *I_nxn_* is the identity matrix and *R* is the residual variance–covariance matrix.

The models were executed in the BGLR package version 1.1.0 in R [36] with 10,000 Monte Carlo chain iterations, without considering the first 5000 and with the thinning of 10.

A 50-fold cross-validation was performed. In each fold, the data were randomly divided into two subsets, one with 70% of the genotypes (training set) and the other with 30% (testing set). In each fold of cross-validation, the mean square of the error (MSE) and the prediction accuracy were calculated, the latter understood as the Pearson coefficient correlation between the predicted and the real breeding values.

The statistical differences in prediction accuracy between the models were based on Welch’s F test. The *p*-values were fitted using the Games–Howell procedure with the ggstatsplot package version 0.11.1 in R [37].

To determine the impact of marker density on the prediction accuracy, random distributed marker subsets of varied sizes (1000, 2000, 3000, 4000, and 5000) were selected for comparison with the full marker collection. The accuracy for each marker density was calculated using the average of the 50-fold cross validation.

## 5. Conclusions

SHW is confirmed to be a valuable genetic resource that can be used to improve genetic resistance to foliar diseases, particularly the three diseases targeted in this study. Models that include genomic information have a higher prediction accuracy than the traditional model, which only includes the pedigree. To improve the prediction accuracy of multi-trait models for resistance to tan spot, spot blotch and Septoria nodorum blotch, other traits must be incorporated with the severity of the diseases in question.

## Figures and Tables

**Figure 1 ijms-24-10506-f001:**
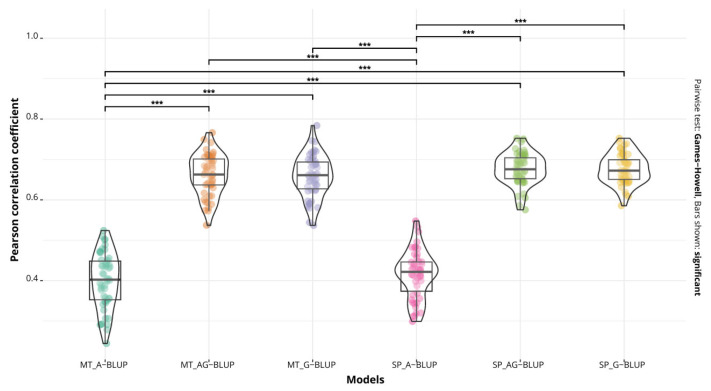
Predictive accuracy of the models A-BLUP, A+G BLUP and G-BLUP, implemented as multi-trait (MT) and single-trait (SP) models for tan spot. Bars show significant differences between the models (*** *p* < 0.01).

**Figure 2 ijms-24-10506-f002:**
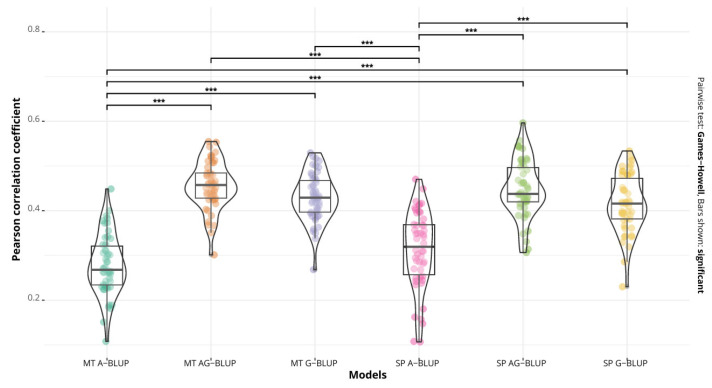
Predictive accuracy of the models A-BLUP, A+G BLUP and G-BLUP, implemented as multi-trait (MT) and single-trait (SP) models for spot blotch. Bars show significant differences between the models (*** *p* < 0.01).

**Figure 3 ijms-24-10506-f003:**
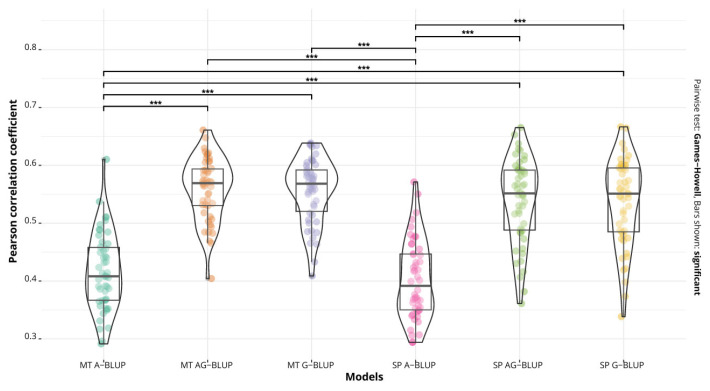
Predictive accuracy of the models A-BLUP, A+G BLUP and G-BLUP, implemented as multi-trait (MT) and single-trait (SP) models for Septoria nodorum blotch. Bars show significant differences between the models (*** *p* < 0.01).

**Figure 4 ijms-24-10506-f004:**
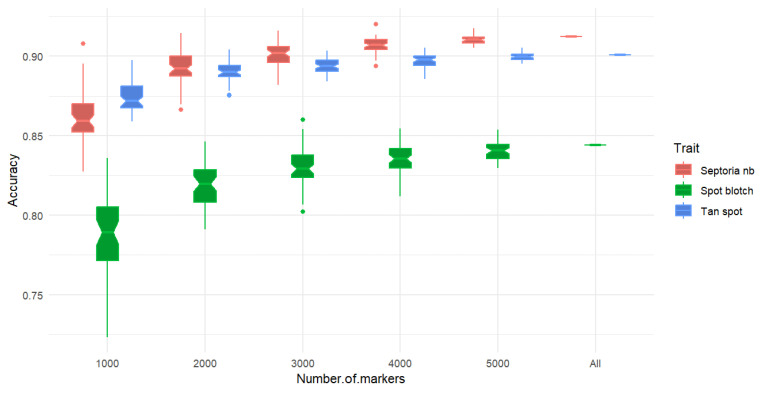
Single prediction accuracy using six marker densities. Plot shows a similar response for the three diseases.

**Table 1 ijms-24-10506-t001:** Descriptive statistics of the BLUE for the severity of tan spot, spot blotch and Septoria nodorum blotch.

Disease	Mean ± SD	Median	Minimum	Maximum	SEM	H^2^	Genomic Heritability
Tan spot	1.63 ± 0.50	1.54	0.92	3.47	0.02	0.88	0.62
Spot blotch	1.64 ± 0.53	1.49	0.99	4.08	0.02	0.84	0.54
Septoria nodorum blotch	1.61 ± 0.57	1.42	0.91	3.67	0.02	0.89	0.68

SD: standard deviation, SEM: standard error of the mean; H^2^: heritability in the broad sense.

**Table 2 ijms-24-10506-t002:** Prediction accuracy for resistance to tan spot, spot blotch and Septoria nodorum blotch based on single-trait and multi-trait models.

		Single-Trait Models		Multi-Trait Models	
Disease	Model	Accuracy	MSE	Accuracy	MSE
Tan spot	A-BLUP	0.41	0.208	0.39	0.224
	G-BLUP	0.67	0.135	0.65	0.151
	A+G BLUP	0.67	0.136	0.66	0.150
Spot blotch	A-BLUP	0.31	0.258	0.27	0.260
	G-BLUP	0.41	0.235	0.42	0.231
	A+G BLUP	0.44	0.228	0.45	0.225
Septoria nodorum blotch	A-BLUP	0.40	0.290	0.41	0.278
	G-BLUP	0.53	0.242	0.55	0.228
	A+G BLUP	0.53	0.241	0.55	0.227

MSE: mean square error.

**Table 3 ijms-24-10506-t003:** Fungal isolates used and their origins, culture media and inoculum concentrations.

Pathogen	Isolate	Origin	Culture Medium	Concentration (Conidia mL^−1^)
*P. tritici-repentis*	CIMFU 531-Ptr1 (race 1)	Yanhuitlan, Oaxaca, Mexico	V8-PDA	4 × 10^3^
*B. sorokiniana*	CIMFU 483 (BSG40M2)	Agua Fría, Puebla, Mexico	30% V8-PDA	7.5 × 10^3^
*S. nodorum*	CIMFU-463 Sn4	Tlanepantla, State of Mexico, Mexico	V8-glucose	1 × 10^7^

## Data Availability

The phenotypic, genomic (G) and pedigree (Amat) data used in this study can be downloaded from the following link https://hdl.handle.net/11529/10548916 (accessed on 29 September 2022).

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
