# Peer review of "Genomic Prediction of Resistance to Tan Spot, Spot Blotch and Septoria Nodorum Blotch in Synthetic Hexaploid Wheat"

_ijms, 2023, doi:10.3390/ijms241310506_

Round 1
Reviewer 1 Report
The work by García-Barrios et al. explored single- and multi-trait genomic prediction (GP) across 400 synthetic hybrid wheat (SHW) lines for three types of leaf-spotting biotic stresses by inputting resistance 1-5 scale and 6,548 DaRT-based SNPs into A-BLUP, G-BLUP and (A+G)-BLUP models with 50-fold CV. Accuracy and MSE scores were within the expected ranges. The A-BLUP model was always outperformed. The work is well framed and addressed key questions in the discipline, named the utility of combined pedigree-based and marker-based prediction models. Furthermore, García-Barrios et al. did manage to calibrate a comprehensive GP tool that certainly will prove useful for predictive breeding of biotic stresses, an often-neglected selection target.
Key pros of the study are that this is a statistically well-designed calibration trial, which at first glance pushes forward the applicability of GP into novel traits. However, a more careful read would invite authors to be more careful at clearly drawing the line (i.e., research gap) at the Introduction (L66) and the Discussion sections on what novelty this report conveys beyond the study system (i.e., wheat and the three leaf-spotting diseases).
Furthermore, because some of the studied traits likely exhibit a strong environmental interaction (e.g., pathogen pressure, and precipitation and humidity likely to predispose leaves), authors must refer from the very begging to previous quantifications of the heritability or additive genetic variance. A general recommendation is to first comment on these values (perhaps in paragraph at L44) before moving into a full genomic prediction study, particularly for traits with a substantial environmental component. For comparison purposes on missing heritability, please also report the genomic-estimated heritability scores within the main figures.
From a more analytical perspective, methods, results and data interpretation are generally coherent, and conclusions are well supported. I am glad that the GP techniques implemented by the authors combined pedigree- and marker-based estimates. Since the latter are know to offer more sensitivity while better controlling for LD, they super-passed the former. Despite this, authors must bring clearer considerations regarding overall LD patterns in the studied SHW panel, the samples’ co-ancestry, the potential preselection of SNP markers from previous GWAS results to harness GP predictive ability (as recommended in Heredity 2016 116:395-408), and the calibration of the training/testing ratio for GP. In this sense, the following key figures are missing: (1) overall LD and kinship profiles, (2) GP calibration curves for the training/testing ratio, and (3) predictive ability as a function of increasing subsets of most predictive SNPs to optimize the minimum number of markers that reach a plateau in the predictive ability. Authors should incorporate these figures as main diagrams.
Please allow me to deepen on the previous points, arguments that authors should embrace more thoroughly in their introductory and Discussion sections. As it is already known, GP works either on the basis of shared relatedness (typically measured as relationships due to recent co-ancestry) or on the basis of linkage disequilibrium (LD) between SNP marker loci and the genetic variants that underlie phenotypic variation (refer to and cite PLoS One 2020 15:e0232201). In the absence of demonstrated relationships or LD in the SHW panel (hard to tell precisely because authors do not present any information on LD among the genotyped samples), there is no apparent basis for the reported predictive accuracy of the models under the Results section, raising the question of whether the reported outcomes could be an statistical artifact. That is way the genetic relationships between training and testing datasets are critically important (and therefore must be optimized, as requested in the aforementioned comment). These insights should be incorporated in the introductory section after L57. Overall LD pattern is neither a difficult calculation. Presenting the distribution of overall pairwise among SNP loci would provide readers essential information to understand and evaluate the underlying architecture for GP. Also relevant could be a LD heat map for most predictive SNPs, which would be helpful to check whether GP-informative SNP markers correlate with low recombination rate and extensive LD, as expected due to physical linkage, shared demography, reduced effective population size and increased drift in low-recombining regions (see Nat Rev Genet 2017 18(2):87-100), besides gene-gene interactions within the same pathway and obviously epistasis.
Related with my point on marker optimization, I must confess that authors did not use more appealing approaches to optimize the SNP dataset for GP in parallel to the 50-fold CV effort. Authors should explore alternative strategies in which they choose the most predictive SNP dataset after a priori GWAS analysis. As a first step they could try a somehow unconventional step in which preliminary GWAS ranking is carried out in the same set of individuals than the one used to develop genomic prediction models (like implemented in soybean by Ravelombola et al. 2019 and 2020). Yet, the predictive ability of these models may be biased (see Nat Rev Genet 2013 14:507-515), besides the fact that the polygenic infinitesimal model that makes GP so unique is lost because relying only on GWAS-derived SNP markers disregards SNPs with lower effects that are usually missed by GWAS approaches but that as a whole contribute accounting for the missing heritability in the latter. Therefore, better approaches to be implemented and reported by the authors are: (1) weighted GP models using GWAS estimates gathered from other (or even the same) panel as part of other studies (see and cite see Heredity 2016 116(4):395-408, although perhaps not available for the studied diseases), and (2) optimization of the marker set by computing saturating curves of the predictive ability given various sets of markers ranked by their predictive ability from the very same GP model and not from parallel GWAS model (see Fig. 1 in New Phytol 2012 194:116–28, and prepare one alike).
Another point that authors should explicitly bring into the Discussion (or a potential new Perspectives section, after L282) is whether marker set optimization for GP would make sense in the present case study, but for that the calibration is needed. Marker optimization would traditionally enable more high-throughput screenings (e.g. Kaspr). Unless authors are planning to use well-validated and economic SNP arrays, marker optimization would pointless for their own breeding pipeline. Still, breeders and nurseries around the world could find an optimize set of markers more useful (instead of the full panel of 6,548 DaRT-derived SNP markers that authors utilized). Any alternative should be explicitly discussed.
In terms of content, I encourage authors to discuss the statistical power explicitly in the Discussion section by adding a paragraph on possible caveats (after L282).
Also to be included at the discussion section (before L283) is a short reflection on whether the studied biotic stress resistance traits have not only been understood in wheat as polygenic, but originally believed to be inherited via few Mendelian loci of major effects, motivating a more compact marker assisted selection (MAS) alternative. After all, MAS is already well in place at nurseries around the world for resistance to biotic pressures at early life stages. Although I recognize that GP has until now played a role more as a ‘proof of concept’ than a factual utilization within breeding programs targeting pathogen resistance, that is not truth for MAS. Ultimately, this is a much valid debate on the GS’s efficiency over MAS’s in a polygenic vs. Mendelian context, which readers will certainly find enriching.
Finally, readability of the work will improve if authors utilized more detailed paragraphs and sections. So far they are understandable, yet miss several methodological, analytical and argumentative details likely due to their very condensed structure. Please ensure a more complete contextualization in the introduction, and debate within the discussion.
Author Response
RESPONSE REVIEWER 1
Comments and Suggestions for Authors by Reviewer 1
RESPONSE: Many thanks for the extensive and efficient review. On the revised manuscripts all your suggestions and comments were added and are highlighted in YELLOW. Revision made based on Reviewer 2 are highlighted in GREEN.
The work by García-Barrios et al. explored single- and multi-trait genomic prediction (GP) across 400 synthetic hybrid wheat (SHW) lines for three types of leaf-spotting biotic stresses by inputting resistance 1-5 scale and 6,548 DaRT-based SNPs into A-BLUP, G-BLUP and (A+G)-BLUP models with 50-fold CV. Accuracy and MSE scores were within the expected ranges. The A-BLUP model was always outperformed. The work is well framed and addressed key questions in the discipline, named the utility of combined pedigree-based and marker-based prediction models. Furthermore, García-Barrios et al. did manage to calibrate a comprehensive GP tool that certainly will prove useful for predictive breeding of biotic stresses, an often-neglected selection target.
Key pros of the study are that this is a statistically well-designed calibration trial, which at first glance pushes forward the applicability of GP into novel traits. However, a more careful read would invite authors to be more careful at clearly drawing the line (i.e., research gap) at the Introduction (L66) and the Discussion sections on what novelty this report conveys beyond the study system (i.e., wheat and the three leaf-spotting diseases).
Furthermore, because some of the studied traits likely exhibit a strong environmental interaction (e.g., pathogen pressure, and precipitation and humidity likely to predispose leaves), authors must refer from the very begging to previous quantifications of the heritability or additive genetic variance. A general recommendation is to first comment on these values (perhaps in paragraph at L44) before moving into a full genomic prediction study, particularly for traits with a substantial environmental component. For comparison purposes on missing heritability, please also report the genomic-estimated heritability scores within the main figures.
RESPONSE: We added the analysis of genomic heritability. This is found in Table 1.
From a more analytical perspective, methods, results and data interpretation are generally coherent, and conclusions are well supported. I am glad that the GP techniques implemented by the authors combined pedigree- and marker-based estimates. Since the latter are know to offer more sensitivity while better controlling for LD, they super-passed the former. Despite this, authors must bring clearer considerations regarding overall LD patterns in the studied SHW panel, the samples’ co-ancestry, the potential preselection of SNP markers from previous GWAS results to harness GP predictive ability (as recommended in Heredity 2016 116:395-408), and the calibration of the training/testing ratio for GP. In this sense, the following key figures are missing: (1) overall LD and kinship profiles, (2) GP calibration curves for the training/testing ratio, and (3) predictive ability as a function of increasing subsets of most predictive SNPs to optimize the minimum number of markers that reach a plateau in the predictive ability. Authors should incorporate these figures as main diagrams.
Please allow me to deepen on the previous points, arguments that authors should embrace more thoroughly in their introductory and Discussion sections. As it is already known, GP works either on the basis of shared relatedness (typically measured as relationships due to recent co-ancestry) or on the basis of linkage disequilibrium (LD) between SNP marker loci and the genetic variants that underlie phenotypic variation (refer to and cite PLoS One 2020 15:e0232201). In the absence of demonstrated relationships or LD in the SHW panel (hard to tell precisely because authors do not present any information on LD among the genotyped samples), there is no apparent basis for the reported predictive accuracy of the models under the Results section, raising the question of whether the reported outcomes could be an statistical artifact. That is way the genetic relationships between training and testing datasets are critically important (and therefore must be optimized, as requested in the aforementioned comment). These insights should be incorporated in the introductory section after L57. Overall LD pattern is neither a difficult calculation. Presenting the distribution of overall pairwise among SNP loci would provide readers essential information to understand and evaluate the underlying architecture for GP. Also relevant could be a LD heat map for most predictive SNPs, which would be helpful to check whether GP-informative SNP markers correlate with low recombination rate and extensive LD, as expected due to physical linkage, shared demography, reduced effective population size and increased drift in low-recombining regions (see Nat Rev Genet 2017 18(2):87-100), besides gene-gene interactions within the same pathway and obviously epistasis.
Related with my point on marker optimization, I must confess that authors did not use more appealing approaches to optimize the SNP dataset for GP in parallel to the 50-fold CV effort. Authors should explore alternative strategies in which they choose the most predictive SNP dataset after a priori GWAS analysis. As a first step they could try a somehow unconventional step in which preliminary GWAS ranking is carried out in the same set of individuals than the one used to develop genomic prediction models (like implemented in soybean by Ravelombola et al. 2019 and 2020). Yet, the predictive ability of these models may be biased (see Nat Rev Genet 2013 14:507-515), besides the fact that the polygenic infinitesimal model that makes GP so unique is lost because relying only on GWAS-derived SNP markers disregards SNPs with lower effects that are usually missed by GWAS approaches but that as a whole contribute accounting for the missing heritability in the latter. Therefore, better approaches to be implemented and reported by the authors are: (1) weighted GP models using GWAS estimates gathered from other (or even the same) panel as part of other studies (see and cite see Heredity 2016 116(4):395-408, although perhaps not available for the studied diseases), and (2) optimization of the marker set by computing saturating curves of the predictive ability given various sets of markers ranked by their predictive ability from the very same GP model and not from parallel GWAS model (see Fig. 1 in New Phytol 2012 194:116–28, and prepare one alike).
RESPONSE: We've added an analysis of the effect of marker density on prediction accuracy. Different densities of markers were tested. Starting at 1000 markers with increments of 1000 until all markers are used (See Figure 4).
We decided not to use GWAS SNP markers as fixed effects for genomic prediction. There are two previous studies for this synthetic wheat panel (See references of Ramirez – Lozano for GWAS in GENES journal of MDPI) , and the QTLs for Tan Spot and Spot Blotch have too small effects to be considered substantial in the models.
In terms of content, I encourage authors to discuss the statistical power explicitly in the Discussion section by adding a paragraph on possible caveats (after L282).
RESPONSE: See additions on lines 196-209 of the DISCUSSION Section on the revised version.
Also to be included at the discussion section (before L283) is a short reflection on whether the studied biotic stress resistance traits have not only been understood in wheat as polygenic, but originally believed to be inherited via few Mendelian loci of major effects, motivating a more compact marker assisted selection (MAS) alternative. After all, MAS is already well in place at nurseries around the world for resistance to biotic pressures at early life stages. Although I recognize that GP has until now played a role more as a ‘proof of concept’ than a factual utilization within breeding programs targeting pathogen resistance, that is not truth for MAS. Ultimately, this is a much valid debate on the GS’s efficiency over MAS’s in a polygenic vs. Mendelian context, which readers will certainly find enriching.
RESPONSE: We have decided to strengthen the introduction (L48-L66) to clarify the advantage of genomic prediction over MAS for disease resistance.
Finally, readability of the work will improve if authors utilized more detailed paragraphs and sections. So far they are understandable, yet miss several methodological, analytical and argumentative details likely due to their very condensed structure. Please ensure a more complete contextualization in the introduction, and debate within the discussion.
RESPONSE: We have tried to correct as much as possible the text following the reviewers’ suggestions. Professional English editor revised the entire manuscript

Reviewer 2 Report
1). Manuscript ID: IJMS-2422155
2). Manuscript Title: Genomic Prediction of Resistance to Tan Spot, Spot Blotch and Septoria Nodorum Blotch on Synthetic Hexaploid Wheat
3). Specific Remarks:
Line 3: Change to “Synthetic Hexaploid Wheat”.
Lines 107 to 109: How were the percentages chosen for selecting SNP markers? Please explain.
--Please follow the Journal format while revising the manuscript.
--Add scientific authority at the end of binomial names of all species when they are mentioned for the first time in the manuscript.
--Include full forms of all abbreviations/acronyms mentioned in the manuscript.
Author Response
RESPONSE TO REVIEWER 2 Comments and Suggestions for Authors by Reviewer 2
RESPONSE: thanks for your revision. Additions based on your comments are highlighted in GREEN. Revisions based on Reviewer 1 are highlighted in YELLOW.
Line 3: Change to “Synthetic Hexaploid Wheat”.
RESPONSE: Corrected
Lines 107 to 109: How were the percentages chosen for selecting SNP markers? Please explain.
RESPONSE: We corrected the percentage of MAF (the correct value is 5%), this value has become standard for genomic prediction studies and GWAS. https://doi.org/10.1186/1471-2164-15-740 https://doi.org/10.1534/g3.113.008227
An example with a filter of 40% missing data for markers is: https://doi.org/10.3389/fpls.2020.00197
--Please follow the Journal format while revising the manuscript.
RESPONSE: Yes. Thanks.
--Add scientific authority at the end of binomial names of all species when they are mentioned for the first time in the manuscript.
RESPONSE: Yes. Done.
--Include full forms of all abbreviations/acronyms mentioned in the manuscript.
RESPONSE: Corrected
